# ADAPTING TO CONTINUOUSLY SHIFTING DOMAINS

**Andreea Bobu, Eric Tzeng, Judy Hoffman, Trevor Darrell**
University of California, Berkeley
`{abobu,etzeng,jhoffman,trevor}@eecs.berkeley.edu`

## 1 INTRODUCTION

Imagine a self-driving car with a recognition system trained in mostly sunny weather conditions. Gradually, it starts to rain, and the self-driving agent must adapt to this change and continue to navigate the roads safely. We think of this weather change as a domain shift (Gretton et al., 2009) from a source domain, sunny weather, to a target domain, rainy weather. The typical supervised learning solution to this problem is to further fine-tune the recognition model on labeled datasets of the target, rainy, domain. However, these labels are often unavailable and it can be prohibitively difficult or expensive to obtain enough labeled data to properly fine-tune the large number of parameters employed by deep, multilayer networks. As such, we would like the network to adapt to the new domain in an unsupervised manner, without any need for labeled target data.

Domain adaptation methods mitigate the harmful effects of domain shift by learning transformations that map the labeled source and the unlabeled target domains to a common embedding. This mapping is often achieved by optimizing the representation to minimize some measure of domain shift, such as maximum mean discrepancy (Tzeng et al., 2014; Long & Wang, 2015) or correlation distances (Sun & Saenko, 2016). More recently, adversarial approaches minimize the discrepancy between domains by training a generator to fool a discriminator by producing transformed source images that are indistinguishable from target images (Ganin et al., 2015; Tzeng et al., 2017).

Although these methods transfer well between similar domains, they produce poor results when the covariate shift is too large (Wulfmeier et al., 2017). We posit that, in many scenarios, domains vary continuously and the shift cannot be effectively captured by two static domains. Instead, we adapt iteratively from one source to many gradually shifted target domains by exploiting the continuity in the shift. A notable line of work is that of continuous manifold learning (Hoffman et al., 2014), where they adapt to evolving visual domains by learning a sequence of transformations on a fixed source representation. One issue that arises from this continuous adaptation procedure is *catastrophic forgetting* (Ratcliff, 1990) – a neural network's general tendency to forget past knowledge as it specializes its weights to the current domain. Our method corrects this issue by ensuring that at every adaptation stage the model continues to consistently classify previously seen examples. Thus, a single model can perform continuous adaptation while maintaining strong performance across all domains.

## 2 CONTINUOUS UNSUPERVISED ADAPTATION

In continuous adaptation, we are presented with a source domain $S$, and multiple target domains $T_i$ that represent continuous shifts of $S$ at time $i$. We assume access to source images $X_s$ and labels $Y_s$ drawn from a source domain distribution $p_s(x, y)$, as well as target images $X_{t_i}$ drawn from target distributions $p_{t_i}(x, y)$, with no labeled observations. We additionally assume that the source domain is similar to the target domain at time $t_0$, that the target domain is smoothly varying, and that $p_{t_0}$ is more similar to $p_s$ than $p_{t_1}$ is to $p_s$. Since direct supervised learning on the target domains is not possible, continuous adaptation instead learns a source representation mapping, $M_s$, and a source classifier, $C_s$, and then adapts that model for use in the stream of target domains.

We present a general framework for continuous adaptation with replay, where we evolve the model to the new distribution while simultaneously guiding it to not deviate too far from how it previously performed on prior distributions. Figure 1 illustrates the structure of the proposed replay model. It is comprised of two major components: sequential unsupervised adaptation to adapt models to new domains, described in Section 2.1, and continuous replay adaptation to maintain performance on previously seen domains, described in Section 2.2

### 2.1 Sequential Unsupervised Adaptation

We introduce an adaptation model that progressively evolves to correctly classify multiple shifted domains. Standard unsupervised adaptation effectively adapts between a single source distribution $p_s(x, y)$ and a single target distribution $p_t(x, y)$ by aligning features from both domains. In other words, they learn the source and target mappings, $M_s$ and $M_t$, so as to minimize the distance between the empirical source and target mapping distributions:

$$M_t \leftarrow \arg\min_{M_t} d(M_s(X_s), M_t(X_t)). \quad (1)$$

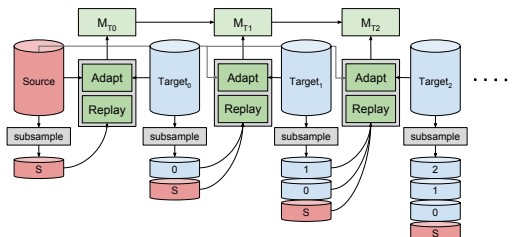

Figure 1: Proposed continuous replay model. At each stage, we save part of the adaptation predictions and use them as "soft" labels for the current domain. We enforce these past soft labels to be matched using a replay loss.

Our method is general and any distance function $d$ can be used. Common choices in recent works include the Kullback-Leibler divergence (Yang et al., 2012), Maximum Mean Discrepancy (MMD) (Gretton et al., 2008), correlation alignment (Sun & Saenko, 2016), and adversarial loss (Liu & Tuzel, 2016; Tzeng et al., 2015; 2017).

When the distance between distributions is minimized, the source classification model, $C_s$, may be directly applied to the target representation; we can, thus, denote both as $C$, and eliminate the need to learn a separate target classifier. We can now find $M_s$ and $C$ by optimizing the supervised objective:

$$M_s, C \leftarrow \arg\min_{M_s, C} \mathcal{L}_{cls}(C(M_s(X_s)), Y_s), \quad (2)$$

where $\mathcal{L}_{cls}$ is commonly chosen to be the cross-entropy loss.In the continuous problem statement, we minimize the distance between one source and multiple targets:

$$M_t \leftarrow \arg\min_{M_t} d(M_s(X_s), M_t(\cup_{i=1}^N X_{t_i})). \quad (3)$$

The above mentioned domain alignment methods would simply conglomerate all target domains together and perform single source to single target adaptation. Unfortunately, standard unsupervised adaptation on a batch of target domains produces poor solutions for the posed optimization problem. Our first step towards improvement is adopting, instead, a sequential approach, where at every stage the model adapts to the next target domain. At each stage, $T_i$, we initialize the current target representation, $M_{T_i}$, using the adapted model from the previous stage, $M_{T_{i-1}}$. We then further adapt between the current target domain data, $X_{T_i}$, viewed under the current target model, and the source domain data, $X_s$, viewed under the original source model.

$$M \leftarrow M_{T_{i-1}}; M_{T_i} \leftarrow \arg\min_M d(M_s(X_s), M(X_{T_i})) \quad (4)$$

By continuing this process at every stage, we ensure successful adaptation to the next target domain. However, while staging enables models to more easily adapt, it does not solve catastrophic forgetting.

### 2.2 Continuous Replay Adaptation

We address the issue of forgetting previous domains by saving the scores for a few previously seen examples and introducing a *replay loss*, $\mathcal{L}_{replay}$, to enforce the response to be the same in the current stage model. We use the cross-entropy loss for $\mathcal{L}_{replay}$. This process is illustrated in Figure 1, where every stage's version of the adaptation model produces a mini-dataset with a few selected observations from their specific domain, together with the predicted classification scores.

Thus, $M_t$ can be updated at every stage via a joint optimization of both the sequential unsupervised adaptation update together with the replay objective:

$$M_t \leftarrow \arg\min_{M_t} [d(M_s(X_s), M_t(X_{t_i})) + \lambda \cdot \mathcal{L}_{replay}(C(M_t(X_p)), Y_p)] \quad (5)$$

where $X_p$ and $Y_p$ are the random samples and their predicted scores saved from previous domains, and $\lambda$ is a replay weight that controls how much to optimize for past domain efficiency.

| Method | 0° | 45° | 90° | 135° | 180° | 225° | 270° | 315° | Average (%) |
|---|---|---|---|---|---|---|---|---|---|
| Source | 99.2 | 61.7 | 17.2 | 29.1 | 39.4 | 29.8 | 15.8 | 51.7 | $43.0 \pm 0.8$ |
| ADDA | 80.8 | 70.4 | 20.8 | 28.6 | 42.1 | 40.2 | 23.8 | 41.2 | $43.5 \pm 1.2$ |
| DANN | 98.6 | 64.7 | 19.9 | 28.4 | 41.4 | 32.9 | 24.2 | 67.3 | $47.2 \pm 1.6$ |
| CUA - no replay (Ours) | 51.6 | 15.1 | 32.7 | 38.7 | 30.4 | 27.1 | 73.6 | 96.0 | $45.7 \pm 1.4$ |
| CUA (Ours) | **90.4** | **84.4** | **82.0** | **77.3** | **85.8** | **88.2** | **92.7** | **96.5** | $\mathbf{90.4 \pm 1.6}$ |
| Target Supervised (Oracle) | 96.9 | 96.7 | 96.8 | 97.4 | 96.6 | 96.5 | 96.8 | 96.4 | 97.0 |

Table 1: Rotated MNIST results for various adaptation methods. We evaluate each row on test data at rotations in $45°$ intervals. The last column contains the average over all test rotations.

Algorithm 1 shows the Continuous Unsupervised Adaptation (CUA) procedure, which sequentially adapts to an evolving target distribution while using replay of past examples to retain prior performance. The method initializes a supervised source model using the labeled source data, and subsamples a few examples from the source data as replay data. A parameter $\alpha$ controls the subsampling rate by deciding how large of a fraction of the data to store. For every new target domain, we fit

---

**Algorithm 1** CUA for continuous adaptation.

1: $M_s, C \leftarrow \arg\min_{M_s, C} \mathcal{L}_{cls}(C(M_s(X_s)), Y_s)$
2: $\{X_p, Y_p\} \leftarrow sample(\{X_s, Y_s\}, \alpha)$
3: $M_t \leftarrow M_s$
4: **for** $i \in \{1...N\}$ **do**
5: $\quad M_t \leftarrow \arg\min_{M_t} d(M_s(X_s), M_t(X_{t_i}))$
6: $\quad\quad\quad\quad + \lambda \cdot \mathcal{L}_{replay}(C(M_t(X_p)), Y_p)$
7: $\quad \hat{Y}_{t_i} \leftarrow C(M_t(X_{t_i}))$
8: $\quad \{X_p, Y_p\} \leftarrow \{X_p, Y_p\} \cup sample(\{X_{t_i}, \hat{Y}_{t_i}\}, \alpha)$
9: **end for**

---

a new target representation $M_t$ by adapting with distance metric $d$ and replay loss $\mathcal{L}_{replay}$. Finally, we subsample $\alpha$-rate data from the current target domain together with the predicted classification scores obtained under this stage's model. [1]

## 3 EXPERIMENTS

We evaluate CUA for unsupervised classification adaptation to continuously shifting domains. For our continuous shifts, we consider the setting of MNIST digits being gradually rotated. This setting causes traditional unsupervised adaptation methods to fail when attempting to adapt to all variations together. We compare our model CUA against multiple state-of-the-art unsupervised adaptation methods that perform adaptation to a batch of target domains. Our method significantly outperforms the competing approaches and nearly reaches fully supervised performance.

Our goal is to adapt from unrotated MNIST digits to MNIST digits of various rotations. We designate rotation by $0°$ to be the labeled source domain, and rotations $45°$, $90°$, $135°$, $180°$, $225°$, $270°$, and $315°$ to be unlabeled target domains. We randomly split the training set in half, assigning 30000 images to the source domain. The remaining 30000 images are further split equally between the seven rotations which comprise the target domain variations. We use LeNet (Cun et al., 1990) as our base architecture in all experiments. As our unsupervised domain adaptation method to adapt between sequential domains we choose the recently proposed ADDA method (Tzeng et al., 2017).

In Table 1, we compare the source only classification (no adaptation); two unsupervised adaptation methods ADDA and DANN (Ganin et al., 2015); CUA with no replay; our full CUA method; and the result of supervised training on all domains. All competing methods that do not use our framework fail catastrophically to adapt to the variety of target domains. The source model, DANN, and ADDA have high accuracy when tested on the source domain $0°$, but fail to adapt to domains that are more distinct (i.e. $90°$ and larger rotations). CUA without replay is able to perform remarkably well on the current target domain, but fails when evaluated on past target domains, in other words suffers from catastrophic forgetting. Finally, our full method, CUA, clearly outperforms all other methods, with high accuracy both on the current and on past domains. On average, our method achieves 90.4% accuracy, a larger than 40% raw improvement over the next competing approach and nearing the performance of a fully supervised model.

---

[1] For a full set of references to related work, discussion of the method, and additional experiments, please see the long version of this paper available on arXiv.

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
