# OpenReview forum: "Adapting to Continuously Shifting Domains"
_ICLR.cc/2018/Workshop — Accept_

### Official Review · AnonReviewer2 · 2018-03-05
**Good paper, accept**

**Rating:** 7
**Confidence:** 4

**Review:**

This paper proposed to tackle a continuous domain adaptation problem by an interesting replaying scheme. Unlike the existing source-target adaptation or multi-source domain adaptation, this paper considers the scenario that the target domains continuously shift over time. With no doubt this is a very challenging problem, the proposed solution is reasonable and gives rise to convincing results in the experiments. The replaying scheme is a nice regularization to prevent the adaptation from negative transfer, a desirable property when the target domains drift away slowly.

Equation (4) is a little confusing since the math notations are not conventional. The authors may consider improve it to make the math language match that in the text. An extended version with more details of the implementation and experiments could be appreciated.

---

### Official Review · AnonReviewer1 · 2018-03-07
**Interesting idea for continuous domain adaptation. The paper lacks of some discussion, justifications, precisions and evaluation, but it is understandable in a short workshop paper.**

**Rating:** 8
**Confidence:** 4

**Review:**

Summary:
This paper adresses the problem of continuous transfer learning where the objective is to continuously adapt a model learned from a source task where supervised learning data are available,
to a sequence of target tasks with no labeled data in the training set.
The authors propose a model that aims at moving closer the target data of each task to the source data with respect to a divergence measure, which is relatively classic. The originality here is that they add a supervised classification term in the objective function aiming at classifying correctly a small dataset of instances from previous tasks with their predicted labels (sof labels). In other words, for a target task t, they try try to learn a projection of M_t to minimizing the divergence between data, such that the classifier learned from source data is still able to find the predicted labels of previously seen examples with respect to the new projection M_t.
An experimental study based on the MNIST digits dataset is provided.

As far as I know, the idea to incorporate soft-labeled instances of previous tasks is a continuous of continuous adaptation is new and the method shows interesting results on a particular tasks. Some discussions and precisions are missing  but it is understandable in a short workshop paper.

In the experimental setup, we do not know how the hyper parameters are tuned/fixed  (lambda, alpha), nor how the other baselines were used. So it is unclear if the results are very specific to the studied task or not. An evaluation on another data would have been informative.

The addition of soft-labels can lead to negative transfer if the first predictions are incorrect. So a good adaptation seems to depend on the ordering to tasks. The fact that the tasks arrive according to the increasing degree of rotation would undoubtedly help.
A discussion on hypothesis where the approach can work or can fail would be interesting as well.

Positive points:
-Interesting idea to deal with continuous learning
-paper reports very good results
-simple idea

Negative points:
-the risk of negative transfer can be high, this point is not discussed
-The choice of the soft-labeled data seems purely heuristic, tuning the tradeoff parameter is unclear
-method evaluated on one particular task

---

### Official Review · AnonReviewer3 · 2018-03-09
**A simple and effective method for tackling an important problem**

**Rating:** 7
**Confidence:** 4

**Review:**

The paper presents a training methodology in a domain adaptation scenario where the learner is provided with a labeled source sample and unlabeled target samples at different time intervals, as the target domain is {varying,drifting,shifting} over time.

The proposed sequential learning algorithm is inspired by existing domain adversarial approaches: at every timestamp, it learns a representation that reduces the "distance" between the corresponding target sample and the source sample, but also reuses a small subsample of self-labeled target samples from previous timestamps (to enforce prediction consistency).

If I were reviewing a conference paper, I would ask some questions on algorithm design and parameter tuning choices. But I think the work fit well in a workshop track, as it provides a simple but effective methodology to handle the problem of shifting domain, which occurs in many real-life applications. I stress the word "methodology" as these ideas can be straightforwardly reused in various learning algorithms.

The paper is well written and succeed in presenting a complete work in three pages. That being said, I feel strange about the fact that the bibliography cites only arXiv references, even when the works have been published in journals and conferences (i.e., has been peer reviewed!)

---

### Decision · Program_Chairs · 2018-03-20
**ICLR 2018 Workshop Acceptance Decision**

**Decision:**

Accept

**Comment:**

Congratulations, your paper was accepted to the ICLR workshop.